# Whether academics' job performance makes a difference to burnout and the effect of psychological counselling—comparison of four types of performers

**Miao Lei[1,2], Gazi Mahabubul Alam [ID][2]\*, Karima Bashir[2,3], Gui Pingping[2,4]**

**1** Student Affairs Division, Yancheng Teachers University, Yancheng, Jiangsu, China, **2** Department of Foundation of Education, Faculty of Educational Studies, University Putra Malaysia, Serdang, Selangor, Malaysia, **3** Department of Education, Faculty of Education, Kebbi State University of Science and Technology, Aliero, Kebbi State, Nigeria, **4** Center for Faculty Development, Sias University, Xinzheng, Henan, China

\* gazi.mahabubul@upm.edu.my, gazimalamb@yahoo.com

**Data Availability Statement:** "All data are in the manuscript and/or supporting information files".

**Funding:** The author(s) received no specific funding for this work.

## Abstract

Treating burnout as an independent variable while performance is the dependent variable, earlier studies revealed that job burnout experienced by academics adversely affects how well they perform. Whether performance may contribute to the emergence of burnout is yet to be analyzed—it is an issue investigated in this paper. Readjusting the nature of the variables, this quantitative study adopted group regression and it discovered that the performance of academics instead regulates their burnout without making performance a consequence of burnout—a new dynamic that challenges the earlier assumption. Following this earlier belief, counselling strategy to boost the employees' psyche was deemed to be the main post-measurement tool to deal with the burnout crisis. With respect to both tenets (current and earlier), psychological counselling was treated as a moderating variable to check whether it is important enough in removing the burnout felt by employees so that they subsequently could function better. It is further discovered that although psychological counselling removes employees' burnout to some extent, it failed to transform them into better-functioning people. This study suggests a pre-measurement counselling strategy will ensure academics are competently engaged since ensuring competency is a fundamental aspect of eliminating a job burnout crisis. The sustained competency of employees will eventually prevent burnout and may halt the transmission of a burnout crisis at large—it adds to this study's theoretical contribution to the topic.

## Introduction

The criteria for job burnout and their schemata evolved out of social and historical practices and experiences, but only since the mid-1970s has the subject revealed the factors that are contributing to burnout and subsequently explored burnout's consequential impact [1, 2]. The

**Competing interests:** The authors have declared that no competing interests exist.

mechanism for investigating and explaining the job burnout crisis has only in recent decades been identified and made it possible to improve the management practices and procedures of an organization [3].

The ongoing increased demands of higher education internationally have created pressure to ensure high-quality services are in fact delivered [4], which has subsequently added to the challenges so that effective university management leads to the desired outcomes [5]. Under these circumstances, studies that have examined job burnout issues in higher education have received greater attention since its inception in the early 1990s [1]. However, a specific study to understand the influence of academics' performance on burnout has to date not been conducted; that is the focus of this study. What follows outlines the scope of this paper by understanding the gaps in knowledge in the existing literature on this subject.

## Research gap and scope

Studies such as those by [6–8] explored the reasons that cause academics to experience job burnout. They pointed out that work overloads, lack of motivation, uneven competition, inefficient and imbalanced management procedures, resource constraints, budgeting issues, and continuous social and family pressures are the prime challenges in a job burnout crisis. While [9] did not deny the facts that identified the fundamental causes of job burnout, they instead stated that it was the incompetence of employees which explained why a job burnout crisis emerged.

As well as identifying the sources of job burnout, attempts were made to discover the solution [1, 10]. Despite a number of strategies suggested (such as motivation and counselling, incentives, resource allocation and distribution, balanced distribution of tasks and working hours, job training, balancing between job demands and resources and balancing one's social, family or private life) to address the job burnout crisis, the gravity of the issue has continued to increase.

Hence, concerns may arise that the identified explanations causing job burnout are neither purely precise nor entirely consistent or accurate. Moreover, the resolution strategies that are subsequently suggested seem to be based on the ambiguous explanations for why job burnout develops; they are also based on functional concern yet faults are inevitable within those strategies [11, 12]. Studies such as [13, 14] further argued that there are now a lot more academic and management staff as well as students participating in higher education. However, many of them undeniably lack an appropriate level of competence, motivation, necessary knowledge, and skills to cope with the pressures of university life.

The group that lacks these fundamental requirements continues to grow [15] and they consequently often contribute to a perplexing workplace environment that works best for vested interests [16]. This might be one of the salient reasons causing job burnout crises and especially in the higher education sector. If so, the current strategies suggested to address job burnout may not be effective enough, since higher education institutions should ideally add more precautionary strategies before offering a position, without being overly dependent on post-job acceptance scenarios when the crisis of job burnout becomes evident.

## Research aim and questions

Challenging this belief, the above discussion highlighted the contention that the performance of staff should ideally lead to a better understanding of the issues concerning job burnout in the university, which is yet to be studied scientifically. Hence, whether an obvious difference exists in connection to job burnout between high—and low—performing groups is a key question that will contribute to further refinement of the subject of job burnout.

Keeping this view in mind, this study intends to compare the performance level groups of academic staff in order to: (1) re-explore the possible reasons causing job burnout in the higher education sector; and (2) offer some pragmatic measures applicable to both pre—and during —job in order to remove the burnout crisis. The following questions are devised to get to the heart of the matter in accordance with the research aim and objectives:

1. Does the performance of academics influence their burnout?

2. How can pragmatic measures be put in place to address job burnout?

The study setting will be explained in the method section and will include more explicit information about China. The research strategy will be explained and justified, followed by the results and discussion, implications, limitations, and conclusions.

## Literature review

This section calls for a hypothesis by challenging the existing factors that cause job burnout and its consequences. The concept of job performance and its measurement scale are explored for consideration here, followed by developing an alternative hypothesis.

### Existing influential factors causing job burnout and its consequences

Studies on the subject of job burnout began in the mid-1970s and essentially, they adopted the qualitative method of analysis [17]. Primarily, the narrative and phenomenological analysis approaches of these studies assumed some factors (such as workload, role conflicts, work-family conflict, years of experience, job autonomy and healthy working relationships) were causing job burnout [2, 18, 19].

From the 1980s onward, many studies were conducted using different quantitative analysis models to test the probable hypotheses wherein job burnout was treated as a dependent variable, while the concerned factors identified were labelled as independent variables [1]. Decades later, research explored a co-relationship/causal relationship between several factors (such as work-family conflict, healthy working relationship and personality) and job burnout [20, 21]. Once the causal/co-relationship had been established, studies concentrated on identifying the consequences of job burnout by using both qualitative and quantitative methods [1]. Since then, comparative studies have been predominantly conducted where job burnout was treated as an independent variable and job performance was a dependent variable [22].

Studies compared two key groups (burnout and non-burnout) and set out to prove that job burnout and job performance bear a negative correlation. Despite a longstanding history of following the above-mentioned pathway for job burnout research (Fig 1), concerns, however, arose that job performance was the key factor that might have triggered job burnout.

If the concern mentioned earlier is proven to be a valid one, prior research on the pathways to burnout may have arrived at a completely different answer but nonetheless reached a conclusion that was not fully agreed on. The ongoing research done on the burnout pathway has erroneously established poor workplace performance as a consequence of job burnout [1, 23].

Hence, this study treats job performance as an independent variable, while job burnout is the dependent variable; this has been done to create a new paradigm on job burnout research that heralds a new pathway on this topic (Fig 2)—travelling from the problem (job performance) and to a destination (job burnout). This new research pathway compares between performing and non-performing groups to discover whether job performance influences burnout, while other relevant factors of burnout (marital status, age, years of employment, administrative positions, and university type) remain the same for both groups.

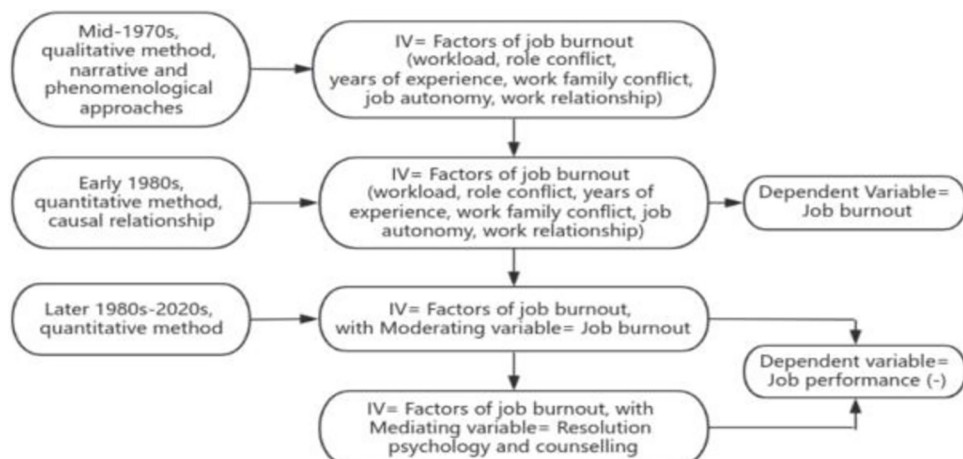

**Fig 1. Flow chart on job burnout research path.** Note. (-) = negative relationship. Source: Authors' creation.

## Job performance: Its measurement and role as a variable

Theoretically, job performance can be defined as an aggregate of employees' behaviors that have some expected value to their workplace (positive or negative). It is essentially the collective expected value of an organization that all employees are envisaged to help achieve [24]. In the context of higher education, [25] emphasized that the collective performance of a university is largely contingent on what the academics in all faculties or departments achieve. Therefore, to enhance how well the academics work, universities not only measure their job performance but also reward them when what they do aligns with the demands of the institution's organizational excellence and individuals' future career prospects [26].

Three major methods have been identified as measuring an employee's job performance. The Annual Compensation Review (ACR), also known as the annual confidential report for some countries, is one of the premier methods of measuring job performance [27]. According to this method, the designated line manager will evaluate how well personnel do their jobs in a descriptive way following what is documented in the job description or duty statement [28].

A longstanding practice of the ACR method suggests that it is in fact an arbitrary process where prejudice, nepotism, favoritism, and playing politics are what really help to achieve a

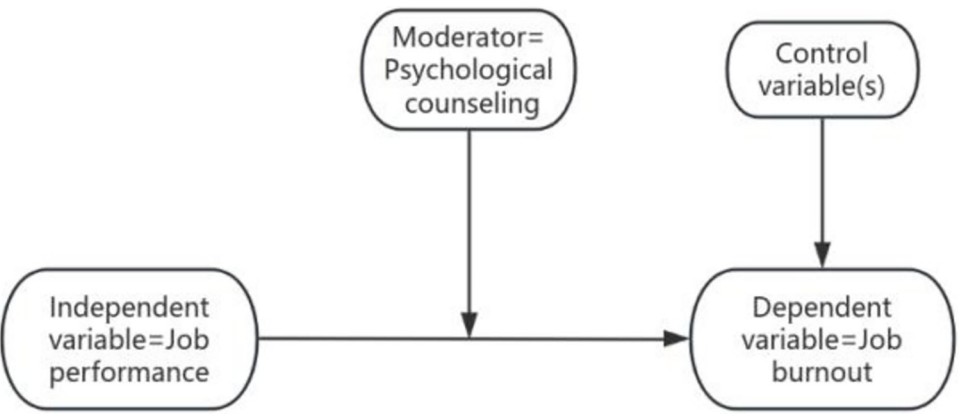

**Fig 2. Proposed research path.** Source: Authors' creation.

higher job performance score instead of the core aspects of the job [27]. To reduce such bad influences, multiple stakeholders were engaged within the ACR to remove these. However, it later failed because it simply contributed to a lobbying and group-based culture in the workplace; hence the ACR became an ineffective tool.

The failure of the ACR model led to the demand for the Individual Work Performance Questionnaire (IWPQ) model [29]. Yet this model also later failed for several reasons: (1) a basic theoretical test of performance may not reflect the practical reality; (2) all employees do not work on the same hierarchical level, thereby the IWPQ model failed to distinguish between certain specified performance expectations and skills; and (3), the IWPQ model failed to identify the specific realities of workers' performance [30].

The incompatibility of both models (ACR and IWPQ) made the Key Performance Indicator (KPI) model popular in recent decades [31]. In order to ensure a more accurate KPI, a 360-degree approach was introduced several years ago [32]. Based on this model, the tasks and responsibilities of an employee are segmented into various micro units that reflect a diversity of differentiated domains. The specified tangible targets identified for each KPI unit are assigned along with the determined score for each employee based on his/her level of hierarchy [33].

Determining the score that goes towards the KPI would normally be maintained through both qualitative and quantitative parameters with the involvement of many stakeholders who work in the system [31]. Despite criticism labelling it as a "numbers game," this KPI has become a globally accepted model for assessing the workplace performance of academics [32] and China is no exception here. Hence, firstly, the score obtained on a KPI is the only official measurement method of evaluating academics' job performance, and secondly, would be used as secondary data.

## Job burnout in higher education: Development of the hypothesis

This section attempts to develop the hypothesis in the light of research questions identified earlier.

In the latest revision of the International Classification of Diseases (ICD-11, 2019), the World Health Organization (WHO) officially designates burnout as an occupational disease, cautioning against its application to describe experiences outside of the workplace. Studies such as [1, 17] defined job burnout from the psychosocial perspective as a chronic stress-related syndrome, one that leads to emotional exhaustion, depersonalization or cynicism, and reduced personal accomplishment, fulfillment or inefficacy. [34] further argued that in severe cases, it could lead to medical conditions such as cardiovascular disease, cardiovascular risk factors, and depression.

Presently, psychological counselling and medical services are the main ways to treat job burnout in many societies but generally they do not adhere to the basic principle known as "prevention is better than cure" [3, 35]. This study therefore intends to hypothesize that job performance fundamentally shapes job burnout by challenging the current hypothesis—job burnout primarily hampers good job performance:

$H_a1$: Job performance has a significant negative influence on job burnout.

$H_a2$: There are significant differences in the role of psychological counselling on job burnout. Psychological counselling moderates job burnout among four different job performance groups.

If these two hypotheses are proven to be right, a strong case would be established to argue that more competent or resilient individuals need to be recruited as a precaution to prevent

the job burnout crisis without overly relying on a post-measurement scheme. This marks a fundamental contribution to the theory of "prevention is better than cure" in job-burnout theory. The following discussions provide a basis that has motivated the authors to devise this hypothesis in higher education by challenging the existing job burnout tenet.

Globally, burnout research was conducted in higher education contexts due to the rising incidents of overloading workplace demands or onerous requests and the insufficient coping competencies of academics [7]. Studies have argued that academics are feeling the strain of various challenging occupational demands, including providing consistently high-quality teaching and supervision to students, the pressure to publish innovative research in high-impact journals, sustaining managerial and entrepreneurial skills and responsibilities, and tenure-related success [8]. Furthermore, the competency of academics has been seriously questioned, which serves as the basis of our hypothesis, as explained in more detail below.

Statistics reveal a surge in the global number of academics from 4 million in 1980 to 13.1 million in 2018 due to the emergence of mass higher education [36]. However, the increase in quantity does not mean that there was a similar increase in the quality of academics, making it impossible to match the standards of elite education [8]. [7] further suggested that academics throughout the world deal with persistent job burnout caused by not having the skills or competency to complete substantial work demands. This perspective will be used in this paper and it has motivated us to devise subsequent hypotheses based on our core hypothesis explained earlier in this sub-section:

$H_a3$: Post-cautionary measures have significant influence on job burnout.

$H_a4$: Pre-cautionary measures have significant influence on job burnout.

The results of these four hypotheses can lead to establishing further findings and discussions for our research questions outlined earlier. Drawing from the model and existing literature, our study employs a quantitative research method with comparative analysis to explore the impact of job performance on job burnout. The following section provides an explanation of the research methods, research context and background information on the population and sample developed for this study.

## Methods

For this study, a quantitative method was adopted. As posited by [37], quantitative research excels in its capacity to provide clear statistical/numerical outcomes and establish relationships among variables.

### Study design

Given that this study seeks to investigate the statistical relationships between academics' performance and job burnout, cross-sectional design is deemed most appropriate for examining any relationships between the variables studied [38].

### Study setting

China is a country in which higher education has undergone rapid expansion since 1999, transitioning from an elite to a mass higher education system [39]. The proliferation of higher education institutions is notable, increasing from approximately 1,000 in 2000 to 2,756 in 2021, reflecting an average annual growth of 100 institutions [40]. The expansion has hugely increased the number of university graduates from 0.83 million in 2000 to 5.054 million in 2021 [40]. Simultaneously, the number of higher education academics surged from 0.46

million in 2000 to 1.87 million in 2021 [40], making China home to one of the largest groups of educators globally.

Given this context, it is important to investigate whether job performance influences burnout in China, as this exploration will help universities recruit more competent academics as a pre-measurement strategy to prevent job burnout. Considering feasibility and generalizability of the findings, this study focuses on Jiangsu province, a major economic hub in China, ranking second nationwide. According to [39], Jiangsu province boasts a total of 167 universities, and it is ranked first in the country. These universities can be categorized into four types, Project 985 universities, Project 211 universities, general undergraduate universities and vocational universities [41].

## Sample/participants

University academics served as the participants in this study. The inclusion criteria involved all academics who have fully participated in the KPI from 2019 to 2023 consecutively. The exclusion criteria were applied to academics who missed the entire KPI or missed a KPI between 2019 and 2023. Hence, academics who have retired or were absent were not included.

To achieve a representative sample the multi-sampling technique was used. Firstly, purposive random sampling was deployed to select one university from each type of university in order to achieve triangulation since the nature of universities varies. Secondly, [42]'s formula for determining sample size: $n = N * Z^2 * P * (1 - P) / e^2 \div [N-1 + Z^2 * P * (1 - P) / e^2]$ was utilized to select the sample out of the total number of academics from the 4 selected universities. Therefore, a sample size of 697 was deemed necessary for a population of 6,328 academics. Subsequently, this sample was proportionally divided using a stratified sampling technique, facilitating the selection of specific respondents from each university. Therefore, 234, 204, 185, and 71 academics were sampled as presented in Table 1.

Finally, the personnel departments of universities sampled assisted in randomly selecting the academics involved in the study based on every third name on the staff list. The randomly selected samples were grouped based on the 4 categorizations of job performance (high, average, low and non-performance), and this will help not just to avoid sample bias but also provide a random visualization of the performance levels of academics. Demographic characteristics of the sample are presented in Table 2.

## Field work and data collection tools

Data was collected from the university archives starting on January 3rd to February 17th, 2024. The data collected includes demographics information, Key Performance Indicator (KPI) results, and psychological counselling data. The use of secondary data obtained from the

**Table 1. Sample size and grouping.**

| University type | Total academics | High P. | Average P. | Low P. | Non-P. | Sampled academics |
|---|---|---|---|---|---|---|
| Project 985 | 2156 | 48 | 94 | 52 | 43 | 237 |
| Project 211 | 1849 | 39 | 79 | 43 | 43 | 204 |
| General undergraduate university | 1680 | 40 | 81 | 32 | 32 | 185 |
| Vocational university | 643 | 16 | 28 | 14 | 13 | 71 |
| Total | 6328 | 143 | 282 | 141 | 131 | 697 |

Note: n = 697.

Source: Authors' own creation

**Table 2. Controlled demographic data.**

| Control Variables | Number | Percentage % |
|---|---|---|
| Gender | | |
| Male | 340 | 49 |
| Female | 357 | 51 |
| Marital status | | |
| Married | 355 | 51 |
| Unmarried | 342 | 49 |
| Age | | |
| 35 or below | 152 | 22 |
| 36–45 | 174 | 25 |
| 46–55 | 185 | 26 |
| 56 or above | 186 | 27 |
| Administrative positions | | |
| Yes | 347 | 50 |
| No | 350 | 50 |
| Years of employment | | |
| 10 or less | 201 | 29 |
| 11–20 | 206 | 30 |
| 21–30 | 201 | 29 |
| 31 or above | 89 | 13 |

Note: n = 697, Extracted by authors

Source: Authors' own creation

personnel department of the sampled university is more reliable than survey responses. Additionally, in order to illuminate multi-year trajectories, the four-year KPI results or scores of academics were compiled.

The demographic data used as controlling variables are presented in Table 2.

In China, the Guiding Opinions on Deepening the Reform of the Assessment and Evaluation System for Universities Teachers [43], mandated universities to establish specific KPIs for measuring each competency and performance of each academic. According to Regulations on the Assessment of Staff in Public Institutions [44], academic performance is generally divided into four categories: high performance (excellent), average performance (qualified), low performance (basically qualified), and non-performance (unqualified). Hence, this grouping will be utilized for the purposes of this study.

The Guiding Opinions on Strengthening Psychological Health Services from [45] stipulated that all higher institutions, should provide psychological health services for staff and regularly provide personnel with psychological assessments and other related services. Hence, data on burnout and psychological counselling was directly obtained from the Mental Health Centre of sampled universities. According to the records results and consultation frequency, burnout and psychological counselling were scored based on a scale. These data provided a visual representation of information on academics who benefit from psychological counselling yearly. The variables, domains and tools are presented in Table 3.

## Ethical considerations

Ethical considerations when undertaking this study were respected. Initially, ethical approval on the safety of the research was obtained from each university before it commenced. Each

**Table 3. Variables, domains and tools.**

| Variables | Domains | Tools/Instruments |
|---|---|---|
| Dependent Variable<br>Burnout | High burnout -3<br>Moderate burnout -2<br>Low burnout -1<br>Non-burnout -0 | Mental Health Centre records |
| Independent Variables | High performance—3<br>Average performance -2<br>Low performance -1<br>Non-performance -0 | KPI |
| Moderating Variable<br>Psychological counselling | Weekly Sessions -3<br>Bi-Weekly Sessions -2<br>Monthly Sessions -1<br>Non-psychological counselling—0 | Mental Health Centre records |

Source: Authors' own creation

university provided an appointment slip which is also locally known as a "conference number". Such an arrangement means that permission has been granted to conduct the study. Selected universities were made aware that participation is voluntary and they can withdraw from the study at any time.

To protect the anonymity of the participants, codes were assigned for all of them numerically in sequential order starting with the first sample selected. Hence, we collected samples from the universities with respondents represented as 1, 2, 3, 4, etc. As such, all respondents' personal information is anonymous and unknown to the researchers. Meanwhile, we designated the universities using letters of the alphabet, A, B, C, and D.

As such, the code for respondents in university A (the first university we collected information from) is A1, A2, etc., to the last respondent. For the second university, we used B1, B2, etc., and pseudonyms were implemented in all the documents as collected from the universities. Before commencing the data collection, the sampled universities were explicitly informed that the data would be used for academic purposes only and confidentiality would be maintained throughout.

## Statistical analysis

To examine the first research question (RQ1) concerning the relationship between job performance and job burnout, a multiple regression analysis was conducted using the latest data. To further explore the relationship between job performance and job burnout and avoid errors caused by sample heterogeneity, group linear regression was employed to investigate the impact of job performance on job burnout among the four job performance groups (Model 1).

A performance group comparison based on burnout in 4 years was mapped out to document the results of RQ1 (Model 2). The study assessed whether job performance affects burnout while controlling for gender, age, and years of employment. For the second research question, hierarchical linear regression is first employed to assess whether psychological counselling significantly influences job burnout and whether it has a moderating effect.

To ascertain the finding in the first step and to avoid errors caused by sample heterogeneity, group regression analysis was conducted among the four performance groups to determine whether psychological counselling has a moderating effect amongst the four job performance groups (Model 1). As well, a performance group comparison based on the role of psychological counselling over four years was presented to support the results of RQ2 (Model 2). Stata

**Table 4. Research questions and statistical methods.**

| Research questions | Hypotheses | Analysis | Methodology |
|---|---|---|---|
| Does the performance of academic staff influence their burnout? | $H_a1$: Job performance has a significant negative influence on job burnout | Linear regression, frequency trends | Quantitative |
| How can pragmatic measures be ensured to address job burnout? | $H_a2$: Psychological counselling moderates job burnout among different job performance groups.<br>$H_a3$: Post-cautionary measures have a significant influence on job burnout.<br>$H_a4$: Pre-cautionary measures have a significant influence on job burnout. | Linear regression, frequency trends | Quantitative |

Source: Authors' own creation

statistical software package version 18 was used for the analysis. The research questions, statistical methods and software are presented in Table 4.

## Results and discussion

In this section the findings and discussion are presented simultaneously. This is followed by an exploration of the conclusion, implications, limitations, and suggestions for further studies in the final section.

### Job-performance: Burnout

The multiple linear regression analysis in Table 5 reveals a significant negative correlation between job performance and job burnout among academics ($\beta$ = -.055, p < .01). Thus, $H_a1$ is supported.

**Table 5. Multiple regression analysis.**

| Variables | Overall |
|---|---|
| Gender | -.035 |
|  | (.063) |
| Marital status | .060 |
|  | (.063) |
| Age | .005 |
|  | (.029) |
| Administrative position | -.045 |
|  | (.063) |
| Years of employment | -.033** |
|  | (.016) |
| Job performance | -.055*** |
|  | (.002) |
| Constant | 5.546*** |
|  | (.172) |
| N | 697 |
| R-squared | .529 |

Note: *** p < .01

** p < .05; n = sample size

Source: Authors' own calculation

**Table 6. Group linear regression analysis.**

| | High | Average | Low | Non |
|---|---|---|---|---|
| Gender | -.006 | .089 | -.192* | -.208* |
| | (.093) | (.075) | (.111) | (.119) |
| Marital status | .002 | -.043 | -.006 | .034 |
| | (.094) | (.075) | (.110) | (.118) |
| Age | .032 | -.033 | .012 | .053 |
| | (.042) | (.033) | (.052) | (.056) |
| Administrative position | -.285** | -.016 | -.051 | .012 |
| | (.093) | (.074) | (.111) | (.118) |
| Years of employment | -.023 | -.020 | -.023 | -.063** |
| | (.024) | (.018) | (.027) | (.032) |
| Job performance | -.056*** | -.157*** | -.296*** | -.100*** |
| | (.015) | (.007) | (.019) | (.010) |
| Constant | 5.810*** | 13.702*** | 21.307*** | 7.380*** |
| | (1.455) | (.547) | (1.220) | (.567) |
| N | 143 | 282 | 141 | 131 |
| R-squared | .156 | .676 | .655 | .476 |

Note: *** $p < .01$

** $p < .05$; n = sample size

Source: Authors' own calculation

To further validate $H_a1$, we conducted a group regression analysis amongst four performance groups to determine whether the level of performance influences job burnout. As presented in Table 6, the results reveal a significant negative correlation between job performance and job burnout across the four job performance groups. In fact, in all the groups, job performance exerts a negative impact on job burnout.

To affirm the negative relationship between job performance and burnout and to complement Model 1's results, i.e. performance determines/influences burnout, the results of the Model 2 analysis reveal that the performance of academics in the high-performance group always remains high. Meanwhile the non-performance group always remains low, as shown in Fig 3.

The results indicate that job performance remains a significant factor influencing job burnout, even after controlling for variables such as gender, age, marital status and so on. Similar to our findings, [6] explained that academics who strive and perform well would endeavor to overcome various stressors in the workplace. Conversely, those who struggle to perform well will eventually suffer burnout due to their inability to cope with the demands of their job. Added to this, in the process of improving performance to the best possible level, academics become more skilled in their tasks and what they accomplish, which has the potential to generate individual benefits (for example, promotion or job recognition) from the organization. These job resources acquired can also alleviate, reduce or prevent job burnout [46].

### Psychological counselling as moderator: potential measurement to address job burnout

Examining the moderating role of psychological counselling on academics with burnout revealed that psychological counselling offers a greater support to remove the burnout crisis. In Block 1, no control variables significantly contribute to burnout. In Block 2, the inclusion of job performance contributes to negatively predicting burnout ($\beta = -.029$, $p < .01$).

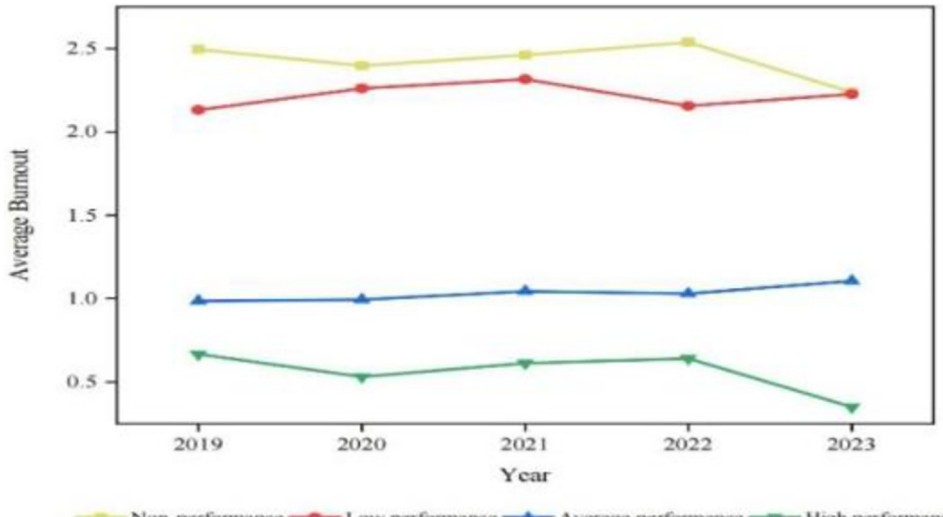

**Fig 3. Job performance group comparison among academics over time (2019 to 2023).** Source: Authors' creation.

In Block 3, what is shown is that psychological counselling negatively influences job burnout ($\beta = -.151$, $p < .05$). When job performance and psychological counselling interact in Block 4, it significantly predicts job burnout ($\beta = -.014$, $p < .01$). Table 7 strongly suggests that

**Table 7. Hierarchical regression analysis.**

|  | Block 1 | Block 2 | Block 3 | Block 4 |
|---|---|---|---|---|
| Gender | -.051 | -.067 | -.049 | -.045 |
|  | (.075) | (.065) | (.075) | (.065) |
| Marital status | -.109 | -.035 | -.107 | -.040 |
|  | (.075) | (.065) | (.074) | (.064) |
| Age | .059 | .031 | .059* | .030 |
|  | (.035) | (.030) | (.035) | (.030) |
| Administrative position | .060 | -.019 | .053 | -.025 |
|  | (.075) | (.065) | (.075) | (.065) |
| Years of employment | -.033 | -.033** | -.032* | -.030* |
|  | (.019) | (.016) | (.019) | (.016) |
| Job performance |  | -.029*** |  | -.022*** |
|  |  | (.002) |  | (.003) |
| psychological counselling |  |  | -.151** | .897*** |
|  |  |  | (.074) | (.319) |
| Job performance *psychological counselling |  |  |  | -.014*** |
|  |  |  |  | (.005) |
| Constant | 2.173*** | 4.154*** | 2.249*** | 3.693*** |
|  | (.108) | (.187) | (.114) | (.242) |
| N | 457 | 457 | 457 | 457 |
| R-squared | .021 | .265 | .030 | .281 |

Note: *** $p < .01$

** $p < .05$; n = sample size

Source: Authors' own calculation

**Table 8. Group regression analysis.**

|  | High | Average | Low | Non |
|---|---|---|---|---|
| Gender | .144 | .115 | -.054 | -.134** |
|  | (.112) | (.092) | (.150) | (.065) |
| Marital status | .051 | -.043 | .041 | -.077 |
|  | (.109) | (.091) | (.149) | (.065) |
| Age | -.049 | -.044 | .104 | .014 |
|  | (.049) | (.041) | (.070) | (.035) |
| Administrative position | -.162 | -.012 | -.051 | -.040 |
|  | (.110) | (.090) | (.153) | (.066) |
| Years of employment | -.034 | -.004 | -.029 | -.018 |
|  | (.029) | (.023) | (.036) | (.017) |
| Job performance | -.061*** | -.142*** | -.272*** | -.098*** |
|  | (.018) | (.008) | (.037) | (.011) |
| Psychological counselling | -.117 | .033 | -2.107*** | -1.575*** |
|  | (.493) | (.093) | (.718) | (.226) |
| Job performance *psychological counselling | .005 | .005 | -.349*** | -.036*** |
|  | (.023) | (.010) | (.104) | (.011) |
| Constant | 6.455** | 12.356*** | 23.399*** | 10.039*** |
|  | (1.957) | (.712) | (1.776) | (.970) |
| N | 87 | 151 | 82 | 53 |
| R-squared | .184 | .710 | .711 | .951 |

Note: *** $p < .01$

** $p < .05$; n = sample size

Source: Authors' own calculation

psychological counselling plays a significant negative moderating role in the relationship between job performance and job burnout among the academics who—as a homogeneous group—are experiencing crises.

Moreover, after excluding academics who did not participate in psychological counselling, the grouped regression analysis presented in Table 8 reveals that psychological counselling has a moderating effect among low (β = -.349, p < .01) and non-performance (β = -.036, p < .01) groups. However, no moderating effect was found in high (β = .005, p>.05) and average (β = .005, p >.05) performance groups. Consequently, the moderating effect of psychological counselling on job burnout varies among the high, average, low and non-performance groups. Based on this finding, $H_a2$ is not supported.

To provide additional insight into the role of psychological counselling in burnout across the four performance groups, longitudinal data analysis was conducted on those groups that received counselling over four years. Fig 4 shows that from 2019 to 2023, the burnout level of those who participate in psychological counselling is lower than those who do not accept psychological counselling.

Likewise, Fig 5 shows that, irrespective of the frequency of psychological counselling and interventions, academics' KPI scores or performance remain consistent. Thus, these findings indicate that psychological counselling will help academics to reduce burnout. In this way $H_a3$ is supported but it has no effect on improving their performance.

In conclusion, as one of the post-measurements, psychological counselling has proven its effectiveness in reducing job burnout. This proves that $H_a3$ is valid. The above results are largely consistent with prior research on using psychological counselling to alleviate job

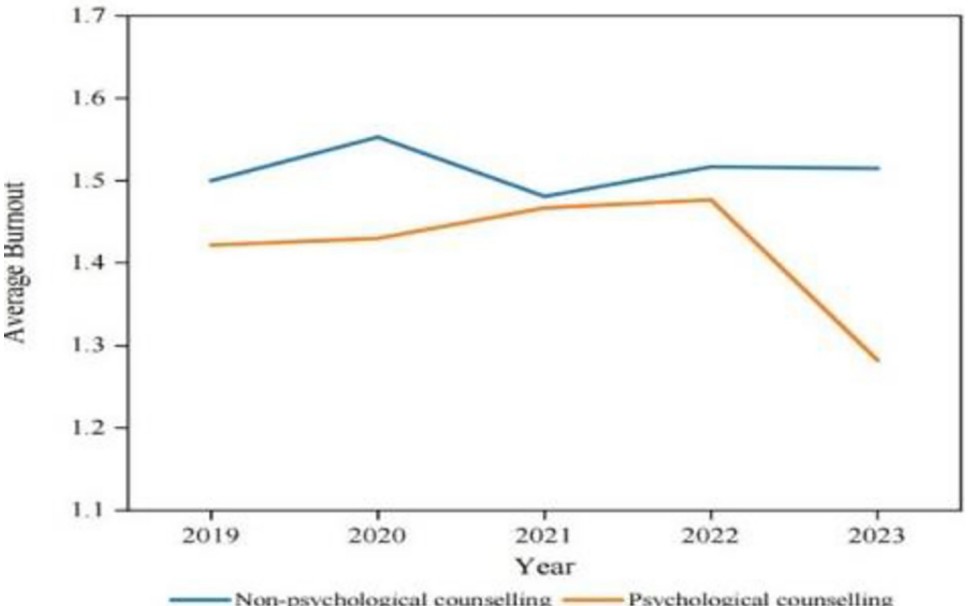

**Fig 4. Burnout comparison based on the role of psychological counselling (2019 to 2023).** Source: Authors' creation.

burnout in other contexts [34, 35]. The findings also offer empirical evidence that psychological counselling has a moderating effect in both low and non-performance groups.

Ideally, psychological counselling has no moderating effect among high and average performance groups. That is because, based on the conclusion reached for research question one, academics who function poorly are more prone to burnout and will seek support from

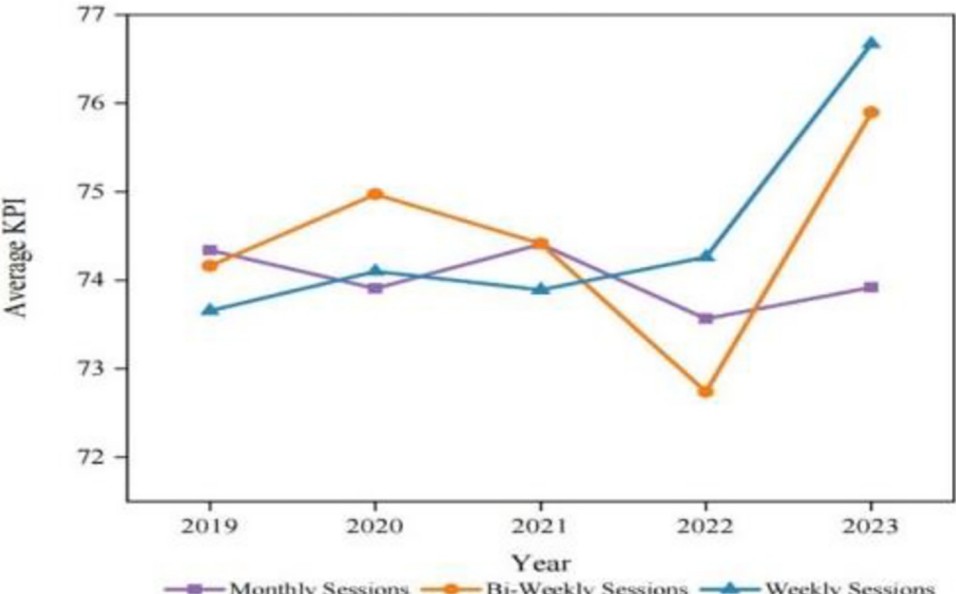

**Fig 5. KPI performance comparison based on the frequency of psychological counselling (2019 to 2023).** Source: Authors' creation.

psychological counselling services. Conversely, those who are able to work well to a high standard do not need the moderating effect of psychological counselling as their burnout level is relatively low. As a result, psychological counselling as a post-measure to alleviate burnout is significant.

These findings indicate the relationship between psychological counselling, burnout, and performance among academics. The observed pattern may indicate that individuals experiencing burnout may engage in a cyclical process, attempting to reduce burnout through psychological counselling. It should be noted that despite the fact that psychological counselling can mitigate burnout, it does not necessarily enhance performance. Hence, the academics may find themselves trapped in a cycle in which an academic's poor performance leads to recurring burnout. Relying on post-measurement for burnout treatment may not be sufficient for permanent resolution and the significance of pre-measurements, as elucidated in $H_a4$ warrants in-depth exploration.

In this context, the confirmation of "the prevention is better than cure" principle within job burnout theory means that $H_a4$ can be supported. To mitigate job burnout amongst academics and enhance their performance, a proactive recruitment system as a pre-measurement strategy should be implemented to avert any signs of a job-burnout crisis. The practical and theoretical implications are discussed next in this paper.

## Theoretical and practical implications

This study significantly contributes to job burnout theory by revealing the relationship between job performance and job burnout amongst academics, yielding valuable practical outcomes. Through expanding job burnout theory and advancing the conventional method from a different perspective, this study makes valid theoretical contributions. While most studies typically consider minimizing employee burnout to improve their performance, this study proposes a new methodological path to exploring the relationship between job performance and burnout.

This study revealed the efficacy of psychological counselling as a post-measurement intervention in reducing burnout among academics. However, it also revealed a limitation wherein such counselling–while it might be effective in reducing burnout—does not improve performance. Since counselling is not ideally meant to improve workplace performance, it means that counselling does not automatically lead to competent, re-energized, motivated, passionate and competent staff. Conversely, a decline in performance may contribute to the persistence of burnout. In light of these findings, it is recommended that human resources recruitment professionals incorporate a comprehensive assessment of all candidates' skills and attributes, i.e. their mental, academic and psychological domains.

Furthermore, in this modern era that has witnessed a dramatic expansion of higher education, many individuals especially in emerging nations wrongly perceive that working as academic means being at liberty to offer less commitment to their workplace [8]. This belief needs to be removed before and after employment commences. As such, in this circumstance, university recruitment systems should incorporate psychological counselling tests to back up competency skills if these institutions want to employ competent, passionate, and self-motived individuals who are ready to work as academics, and can overcome any setbacks.

Applicants should be comprehensively assessed, ensuring their ability to work very hard and diligently, and establishing their psychological fitness by evaluating motivations, passions and determinations relevant to the academic role and required duties or skills. Upon recruitment of the right personnel, arrangements for continuous professional development (both academic and psychological) must be ensured so that academics are able to cope with ongoing changes in the higher education system, especially given the rapid advances in technology.

Moreover, the university management teams should improve training programs that can enhance staff members' job performance by addressing where their skills are lacking and enhancing their mental health.

## Limitations and future studies

Although this study argued that job performance influences burnout, it has the following limitations. Firstly, although this study proposed the inclusion of psychological tests during the recruitment drives, it has not provided the psychological parameters that can be utilized during recruitments tests. Likewise, the findings' generalizability may be limited by the sample originating from only one province in China. It is important to acknowledge the diverse nature of workplaces, industries, geographies, cultural practices, traditions, and contexts given that China is not a homogenous country. To ensure the universality of the findings, future research should include a more diverse and larger sample.

## Conclusions

This study expanded on the job burnout theory to present findings that establish the influence of job performance on burnout as felt by university academics. It verified that performance is one of the causes of job burnout within higher education. While integrating the psychological counselling concept as a moderating variable in the relationship between job performance and burnout, the findings assert that burnout can be resolved through psychological counselling as a temporary remedy. However, this does not mean that a non-performing group can automatically be turned into a performing group. Hence, to prevent job burnout, the correct recruitment strategies should be followed. A key component of this is to include involving a panel of psychologists to assess the cognitive and emotional fitness and resilience of candidates, not just their academic skills and medical screenings.

## Supporting information

**S1 Table. Data for Table 5: Multiple regression analysis.**
(PDF)

**S2 Table. Data for Table 6: Group linear regression analysis.**
(PDF)

**S3 Table. Data for Fig 3: Job performance group comparison among academics over time (2019 to 2023).**
(PDF)

**S4 Table. Data for Table 7: Hierarchical regression analysis.**
(PDF)

**S5 Table. Data for Table 8: Group regression analysis.**
(PDF)

**S6 Table. Data for Fig 4: Burnout comparison based on the role of psychological counselling (2019 to 2023).**
(PDF)

**S7 Table. Data for Fig 5: KPI performance comparison based on the frequency of psychological counselling (2019 to 2023).**
(PDF)

## Acknowledgments

The authors sincerely thank all the sampled universities that voluntarily provided secondary data.

## Author Contributions

**Conceptualization:** Gazi Mahabubul Alam.

**Data curation:** Miao Lei, Karima Bashir.

**Formal analysis:** Miao Lei, Gazi Mahabubul Alam, Karima Bashir, Gui Pingping.

**Methodology:** Gazi Mahabubul Alam.

**Validation:** Karima Bashir.

**Writing – original draft:** Miao Lei, Gazi Mahabubul Alam.

**Writing – review & editing:** Gazi Mahabubul Alam, Karima Bashir.

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
