## [Decision Letter · Decision Letter 0]

13 May 2024

PONE-D-24-07762Whether academics’ job performance makes a difference to burnout and the effect of psychological counselling—comparison of four types of performersPLOS ONE

Dear Dr. Mahabubul Alam,

Thank you for submitting your manuscript to PLOS ONE. After careful consideration, we feel that it has merit but does not fully meet PLOS ONE’s publication criteria as it currently stands. Therefore, we invite you to submit a revised version of the manuscript that addresses the points raised during the review process.

We look forward to receiving your revised manuscript.

Kind regards,

Heba E. El-Gazar

Academic Editor

PLOS ONE

Journal Requirements:

2. Thank you for stating the following in your Competing Interests section: N/A

3. We note that your Data Availability Statement is currently as follows: "May I note that Our research is not a medical one in nature and we only analyzed secondary data in this paper. No human/animal participants were involved either for data collection for trail. We collected necessary secondary data by contacting relevant department from the Universities sampled."

Reviewers' comments:

Reviewer's Responses to Questions

**Comments to the Author**

1. Is the manuscript technically sound, and do the data support the conclusions?

Reviewer #1: Yes

Reviewer #2: Yes

2. Has the statistical analysis been performed appropriately and rigorously? 

Reviewer #1: Yes

Reviewer #2: Yes

3. Have the authors made all data underlying the findings in their manuscript fully available?

Reviewer #1: Yes

Reviewer #2: Yes

4. Is the manuscript presented in an intelligible fashion and written in standard English?

Reviewer #1: Yes

Reviewer #2: Yes

5. Review Comments to the Author

Reviewer #1: Title: Whether academics’ job performance makes a difference to burnout and the effect of psychological counselling—comparison of four types of performers

I have assessed your manuscript and would like to kindly request a clarification:

Abstract

- Keywords: Please consider visiting Mesh to select your keywords. using is very helpful in terms of increasing your article visibility & citation chance! Please Visit Their Website: https://www.ncbi.nlm.nih.gov/mesh/\\

You need to organize the manuscript more properly. There are several recurring paragraphs, such as those related to the history of research into the causes of burnout, which makes the research complicated to read and confusing.

Background

- The study aim: it was very long and make confused. The introduction for the aim section is not necessary. Please mention the study aim directly.

- Fig 2 in page 7 don’t clarify the psychological counseling role as a mediator as you mentioned in the research hypothesize in page 9. Please clarify in the figure.

- I suggest to merge the Research Context section in page 11 with the literature review with avoid repetition. It is too long and hard to read.

Methods

- The study design is not clear. What type of the quantitative method was used? and the used design should be supported by a cited materials that justifies using it.

- Some main missing points: a brief description of the study setting and Chinese universities, population, target universities. I suggest adding separate paragraph for the study setting description.

- The study sample section needs to organize. You mentioned in the beginning you used (a multi-stage sampling approach). Then, you mentioned in the second paragraph (a stratified sampling technique). I suggest it need to arranged in the beginning of the section as, a stratified sampling technique was used by divides the population into smaller groups (strata) based on shared characteristics of the target individual.

- Please clarify the used sample size formula.

- Sample size and grouping were clear enough and organized in table 1.

- Inclusion and exclusion criteria should be mentioned in the sample section not in data collection section.

- The tools of data collection not mentioned. Please describe the used tools in details, clarify if developed by the researcher or adapted from original references, and identify the scoring system used.

Validity

I have read the entire methods section. However, I could not find any details about tools validity. Please describe tools validity in separate section entitled (tools validity).

Reliability

It not mentioned. Please explain the study tools’ reliability.

Field of work

Please explain the method and time of questionnaires distribution.

Statistical analysis

It is clear in table 4.

Results and discussion

I see that the discussion was answered the research hypothesis.

Reviewer #2: 1. Is the manuscript technically sound, and do the data support the conclusions?

The manuscript demonstrates technical soundness through rigorous statistical analysis, systematic methodology, and appropriate data-conclusion alignment, supporting the validity of the research findings.

The research paper conducted group regression analysis among four performance groups to determine the moderating effect of psychological counseling, ensuring a rigorous statistical approach. The study utilized multiple regression analysis to explore the relationship between job performance and burnout, demonstrating a methodical data analysis process.

2. Has the statistical analysis been performed appropriately and rigorously?

The research paper employed group regression analysis among four performance groups to determine the moderating effect of psychological counseling, showcasing a robust statistical approach. Multiple regression analysis was utilized to investigate the relationship between academics' performance and job burnout, indicating a comprehensive statistical analysis methodology. The study justified the use of quantitative research to establish clear statistical outcomes and relationships between variables, highlighting the appropriateness of the statistical methods employed.

3. Have the authors made all data underlying the findings in their manuscript fully available?

The authors have taken measures to maintain data confidentiality, collect comprehensive demographic information, and rely on reliable sources, ensuring the availability of essential data for the study.

Authors have ensured the anonymity of participants by assigning numerical codes sequentially and using pseudonyms for universities, maintaining confidentiality. Data on burnout and psychological counseling were directly obtained from the Mental Health Centers of the sampled universities, ensuring reliable information for the study.

4. Is the manuscript presented in an intelligible fashion and written in standard English?

The manuscript is well-written, free from language errors, and presented in a manner that is easily understandable, meeting the standards of clarity and language proficiency expected in academic publications. It is presented in a clear and intelligible fashion, adhering to standard English conventions throughout the text. The authors have effectively communicated their research findings, hypotheses, and implications in a coherent manner, enhancing the readability and understanding of the study.

6. PLOS authors have the option to publish the peer review history of their article (what does this mean?). If published, this will include your full peer review and any attached files.

Reviewer #1: **Yes: **Heba Moussa Hagrass

Reviewer #2: **Yes: **Hussein M Magdi

---

## [Author Response · Author response to Decision Letter 0]

21 May 2024

Date: May 20, 2024

To

Mr. Heba E. El-Gazar

Academic Editor

Plos one 

San Francisco, California, USA

Subject: Resubmission of the revised manuscript PONE-D-24-07762R1. 

Dear Mr. Heba E. El-Gazar 

We humbly remain grateful to you for appraising our paper and providing your insightful comments on our study. Two esteemed reviewers have reviewed the paper and we remain very grateful to their meticulous inputs that are fundamental in improving this paper. We have addressed the comments of you and both reviewers and they are reported in blue color text. Kindly find below the responses for each comment, and we remain committed towards ensuring we satisfy your requirement for publication.

We appreciate your time and consideration.

Thanking you,

Authors 

Journal Requirements:

Response: Thank you so much. Considering you comment and comment from reviewer 1, we have revised the manuscript following the requirements of PLOS ONE. Thank you.

2.Thank you for stating the following in your Competing Interests section: N/A

Response: Thank you for your kind comment. Thank you. We have requested to add “The authors have declared that no competing interests exist” at cover letter.

3. We note that your Data Availability Statement is currently as follows: "May I note that Our research is not a medical one in nature and we only analyzed secondary data in this paper. No human/animal participants were involved either for data collection for trail. We collected necessary secondary data by contacting relevant department from the Universities sampled."

Please confirm at this time whether or not your submission contains all raw data required to replicate the results of your study. Authors must share the “minimal data set” for their submission. PLOS defines the minimal data set to consist of the data required to replicate all study findings reported in the article, as well as related metadata and methods.

Response: Thank you so much. Following your suggestion, we have submitted all data required to replicate the results of the study. We have also clarified the statement on ‘Availability of data and materials. Please see pages 27, 30, 31 and files S5, S6, S7 and S8 Table and S3, S4 and S5 Fig. Thank you.

Tables 5, 6, 7 and 8 + figures 3, 4 and 5 are through the data collected. The “minimal data set” for each table and figure are provided.

4.Please include captions for your Supporting Information files at the end of your manuscript, and update any in-text citations to match accordingly.

Response: Thank you so much. Following your suggestion, we have included captions for the supporting information files at the end of the manuscript. Please see pages 30, 31 and files S5, S6, S7 and S8 Table and S3, S4 and S5 Fig. Thank you.

5.Please review your reference list to ensure that it is complete and correct. If you have cited papers that have been retracted, please include the rationale for doing so in the manuscript text, or remove these references and replace them with relevant current references. Any changes to the reference list should be mentioned in the rebuttal letter that accompanies your revised manuscript.

Response: Thank you so much. Following your suggestion, we have ensured the reference is complete and correct. After the revisions, we deleted some references and re-numbered the referencing list. Please see pages 27 to 30. Thank you.

Reviewer 1 :

Comment: 

1. Abstract- Keywords: Please consider visiting Mesh to select your keywords. using is very helpful in terms of increasing your article visibility & citation chance! Please Visit Their Website:

https://www.ncbi.nlm.nih.gov/mesh/\\ 

Response: Thank you for the keen comment. We have confirmed and modified the keywords using Mesh on demand. Please see page 1. Thank you.

Comment: 

2. You need to organize the manuscript more properly. There are several recurring paragraphs, such as those related to the history of research into the causes of burnout, which makes the research complicated to read and confusing.

Response: Thank you for the insightful comment. Taking into consideration your comment and comment from the editor, we have organized the manuscript more appropriately. 

We have also reorganized the history of research into the causes of burnout in a more concise manner. Please see page 4, 5 and 6. Thank you.

Comment: 

3. Background-The study aim: it was very long and make confused. The introduction for the aim section is not necessary. Please mention the study aim directly.

Response: Thank you so much. Following your suggestion, we have stated the aim of the research directly. Please see page 3 and 4. Thank you.

Comment: 

4. Background-Fig 2 in page 7 don’t clarify the psychological counseling role as a mediator as you mentioned in the research hypothesize in page 9. Please clarify in the figure. 

Response: Thank you for your profound comment. We have clarified the psychological counseling role as a moderator in Fig 2. Please see page 6. Thank you.

Comment: 

5. Background- I suggest to merge the Research Context section in page 11 with the literature review with avoid repetition. It is too long and hard to read.

Response: Thank you so much. Considering your suggestion to add a brief description of the study setting (comment no. 7), we have reorganized the research context and created a study setting section. Similarly, we have merged some relevant information with literature review. Please see page 11 and 12. Thank you.

Comment: 

6. Methods-The study design is not clear. What type of the quantitative method was used? and the used design should be supported by a cited materials that justifies using it.

Response: Thank you so much. Following your suggestion, we have stated and justified the research design used for the study. Please see page 11. Thank you.

Comment: 

7. Methods-Some main missing points: a brief description of the study setting and Chinese universities, population, target universities. I suggest adding separate paragraph for the study setting description.

Response: Thank you so much. Following your suggestion, we have added the study setting sub-section. Please see page 11 and 12. Thank you.

Comment:

8. Methods-The study sample section needs to organize. You mentioned in the beginning you used (a multi-stage sampling approach). Then, you mentioned in the second paragraph (a stratified sampling technique). I suggest it need to arranged in the beginning of the section as, a stratified sampling technique was used by divides the population into smaller groups (strata) based on shared characteristics of the target individual.

Response: Thank you so much. Following your suggestion, we have reorganized the sample selection and clarified the participants for the study. Please see sub-section ‘sample/participants’, page 12 and 13. Thank you.

Comment: 

9. Methods-Please clarify the used sample size formula. 

Response: Thank you so much. Following your suggestion, we have clarified the sample size formula. Please see page 12. Thank you.

Comment: 

10. Methods-Sample size and grouping were clear enough and organized in table 1.

Response: Thank you for the generous comment. Thank you.

Comment: 

11. Methods-Inclusion and exclusion criteria should be mentioned in the sample section not in data collection section.

Response: Thank you so much. Following your suggestion, we have reorganized the inclusion and exclusion criteria and included it in the sample/participants sub-section. Please see page 12. Thank you.

Comment: 

12. Methods-The tools of data collection not mentioned. Please describe the used tools in details, clarify if developed by the researcher or adapted from original references, and identify the scoring system used. 

Response: Thank you so much. The tools and data collection method have been reorganized in a clearer manner. Additionally, the scoring system used in each variable is shown in Table 3. Please see sub-section ‘Field work and data collection tools’, page 13, 14 and 15. Thank you.

Comment:

13. Validity-I have read the entire methods section. However, I could not find any details about tools validity. Please describe tools validity in separate section entitled (tools validity).

Response: Thank you so much. The study used secondary data, hence there is no requirements for tool validity. Please see page 13. Thank you.

Comment:

14.Reliability-It not mentioned. Please explain the study tools’ reliability.

Response: Thank you so much. The study used secondary data from sampled universities, hence the instruments are reliable and there is no need for reliability test. Please see page 13. Thank you.

Comment:

15.Field of work-Please explain the method and time of questionnaires distribution.

Response: Thank you so much. The method and time of the field work has been explained. Furthermore, the research only analyzed secondary data, hence, there was no questionnaire distribution. Please see sub-section ‘Field work and data collection tools’, page 13. Thank you.

Comment:

16. Statical analysis-It is clear in table 4.

Response: Thank you for the kind comment. Thank you.

Comment:

17. Results and discussion-I see that the discussion was answered the research hypothesis. 

Response: Thank you for the generous comment. Thank you.

Reviewer 2 :

Comment:

1. Is the manuscript technically sound, and do the data support the conclusions?

The manuscript must describe a technically sound piece of scientific research with data that supports the conclusions. Experiments must have been conducted rigorously, with appropriate controls, replication, and sample sizes. The conclusions must be drawn appropriately.

The manuscript demonstrates technical soundness through rigorous statistical analysis, systematic methodology, and appropriate data-conclusion alignment, supporting the validity of the research findings.

The research paper conducted group regression analysis among four performance groups to determine the moderating effect of psychological counseling, ensuring a rigorous statistical approach. The study utilized multiple regression analysis to explore the relationship between job performance and burnout, demonstrating a methodical data analysis process.

Additionally, the manuscript's methodology involved a quantitative approach, ensuring a structured and systematic research design. The data collection procedures were carefully explained, with anonymity measures in place to protect participants' privacy and ensure research integrity. Also, the conclusions drawn in the paper were based on the data presented, with a clear link established between job performance, burnout, and the moderating effect of psychological counseling.

Response: Thank you for the generous comment. Thank you.

Comment:

2. Has the statistical analysis been performed appropriately and rigorously?

The statistical analysis in the research paper appears to have been performed appropriately and rigorously, utilizing various statistical techniques to investigate the research questions effectively.

The research paper employed group regression analysis among four performance groups to determine the moderating effect of psychological counseling, showcasing a robust statistical approach. Multiple regression analysis was utilized to investigate the relationship between academics' performance and job burnout, indicating a comprehensive statistical analysis methodology. The study justified the use of quantitative research to establish clear statistical outcomes and relationships between variables, highlighting the appropriateness of the statistical methods employed.

In addition, a multi-stage sampling approach was implemented to achieve a representative sample, ensuring the statistical analysis was conducted with careful consideration of sample heterogeneity and data accuracy. In turn, the statistical methods used in the research paper align with the study's objectives, providing a rigorous analysis framework to explore the dynamics between job performance, burnout, and the moderating effect of psychological counseling.

Response: Thank you for the kind comment. Thank you.

Comment:

3. Have the authors made all data underlying the findings in their manuscript fully available?

Authors have taken measures to maintain data confidentiality, collect comprehensive demographic information, and rely on reliable sources, ensuring the availability of essential data for the study. 

Authors have ensured the anonymity of participants by assigning numerical codes sequentially and using pseudonyms for universities, maintaining confidentiality. Data on burnout and psychological counseling were directly obtained from the Mental Health Centre of sampled universities, ensuring reliable information for the study.

The collected demographic data includes gender, marital status, age, administrative position, and years of employment, providing a comprehensive dataset for analysis.

While the specific raw data points behind means, medians, and variance measures are not explicitly mentioned, the study's reliance on secondary data from universities' personnel departments enhances the reliability and availability of data for further analysis.

Response: Thank you for the generous comment. Thank you. 

Comment:

4. Is the manuscript presented in an intelligible fashion and written in standard English?

The manuscript is well-written, free from language errors, and presented in a manner that is easily understandable, meeting the standards of clarity and language proficiency expected in academic publications. It is presented in a clear and intelligible fashion, adhering to standard English conventions throughout the text. Authors have effectively communicated their research findings, hypotheses, and implications in a coherent manner, enhancing the readability and understanding of the study.

Response: Thank you for the kind comment. Thank you.

---

## [Editor Report · Decision Letter 1]

31 May 2024

Whether academics’ job performance makes a difference to burnout and the effect of psychologicalcounselling—comparison of four types of performers

PONE-D-24-07762R1

Dear Dr. Alam,

We’re pleased to inform you that your manuscript has been judged scientifically suitable for publication and will be formally accepted for publication once it meets all outstanding technical requirements.

Kind regards,

Heba E. El-Gazar

Academic Editor

PLOS ONE

Additional Editor Comments (optional):

no comment
---

## [Editor Report · Acceptance letter]

5 Jun 2024

PONE-D-24-07762R1 

PLOS ONE

Dear Dr. Alam, 

I'm pleased to inform you that your manuscript has been deemed suitable for publication in PLOS ONE. Congratulations! Your manuscript is now being handed over to our production team.

Kind regards, 

on behalf of

Dr. Heba E. El-Gazar 

Academic Editor

PLOS ONE